# Clopidogrel responder status is uninfluenced by CYP2C19*2 in Danish patients with stroke

Charlotte Lützhøft Rath[1]*, Niklas Rye Jørgensen[2,3], Troels Wienecke[1,4]

1 Neurovascular Centre, Department of Neurology, Zealand University Hospital, Roskilde, Denmark,
2 Department of Clinical Biochemistry, Rigshospitalet, University of Copenhagen, Copenhagen, Denmark,
3 OPEN, Odense Patient data Explorative Network, Odense University Hospital/Institute of Clinical Research, University of Southern Denmark, Odense, Denmark, 4 Department of Clinical Medicine, University of Copenhagen, Copenhagen, Denmark

* clra@regionsjaelland.dk

## Abstract

### Background

Antiplatelet therapy is a cornerstone of secondary stroke prevention, but the responsiveness to antiplatelet medication varies among patients. Clopidogrel is a pro-drug that requires hepatic transformation to reach its active metabolite. Single nucleotide polymorphisms (SNPs) in key enzymes or the target adenosine diphosphate (ADP) receptor on the platelet surface are believed to be involved in clopidogrel-mediated platelet inhibition and decreased antiplatelet effect with high-on-treatment platelet reactivity (HTPR).

### Objective

This study investigated whether specific SNPs in key hepatic enzymes (CYP2C19*2, *3, *17, CYP3A4*1G, and NR1I2) or the ADP receptor (PR2Y12) are associated with HTPR to clopidogrel.

### Patients & methods

This observational study included patients with ischemic stroke (IS) and transient ischemic attacks (TIAs) receiving clopidogrel at a dose of 75 mg/day. Patients were genotyped for eight different SNPs in the genes encoding CYP2C19, CYP3A4, NR1I2, and the P2Y12 receptor.

### Results

Of the 103 patients that were included, 30.7% carried the CYP2C19*2 allele and had higher platelet reaction unit (PRU) values than non-carriers, but no patients showed HTPR. Carriers of the *17 allele had higher platelet inhibition but showed no difference in PRU values compared with non-carriers. The remaining SNPs were neither associated with PRU nor with platelet inhibition.

**Data Availability Statement:** All relevant data are within the manuscript and its Supporting Information files.

**Funding:** T.W. Grant number NNF14OC0012727. Novo Nordisk Foundation. URL: https://

novonordiskfonden.dk/en/ The funders had no role
in study design, data collection and analysis,
decision to publish, or preparation of the
manuscript.

**Competing interests:** The authors have declared
that no competing interests exist.

## Conclusions

Patients with IS and TIAs treated with 75 mg clopidogrel/day do not have HTPR. A genetic analysis of CYP2C19*2, *3, *17, CYP3A4*1G, and NR1I2 revealed no associations with clopidogrel HTPR. CYP2C19*2 carriers and patients with HTPR in the acute phase after ischemic stroke or transient ischemic attacks exhibit higher PRU values, but not long-term treatment HTPR.

## Introduction

Clopidogrel prevents recurrent ischemic stroke (IS) with the same efficacy as aspirin in combination with extended-release dipyridamole [1]. The ex vivo prevalence of high-on-treatment platelet reactivity (HTPR) varies from 8 to 61% in clopidogrel-treated patients with IS and transient ischemic attacks (TIAs) [2]. Furthermore, a recent meta-analysis by Fiolaki et al. showed that the clopidogrel HTPR prevalence is 27%, and that patients with HTPR have an increased risk of recurrent IS and TIAs [3]. In patients with coronary artery disease undergoing percutaneous intervention, tailored antiplatelet therapy (either increased drug dosage or shift of medical therapy), guided by platelet function tests (PFTs), is associated with less occurrence of death and stent thrombosis without an increased risk of bleeding complications or clinical adverse events [4].

Clopidogrel is a pro-drug that requires a two-step oxidation by the cytochrome P450 system. Genetic variants (single nucleotide polymorphisms, SNPs) in key enzymes involved in this process or in the formation of the P2Y12 receptor on the platelet surface might affect phenotypic platelet responses measured with PFTs.

This study investigates whether patients with ischemic stroke during secondary prophylactic treatment with 75 mg clopidogrel daily carrying the SNPs CYP2C19*2, *3, or *17, or CYP3A4*1G, NR1I2, or P2Y12 [5–9] have a higher risk of HTPR, and whether an increased dosage of clopidogrel can overcome HTPR in these patients, as has been shown for patients with cardiovascular disease (CVD) [10].

CYP2C19 is believed to be the most important enzyme in the two-step oxidation of clopidogrel to its active metabolite. The *2 allele is a loss-of-function (LOF) mutation resulting in complete loss of enzymatic activity. Consequently, carriers of the *2 allele show reduced formation of the active clopidogrel metabolite and hence reduced clopidogrel-induced platelet inhibition. The *17 allele is a gain-of-function (GOF) mutation leading to increased enzyme function and hence increased clopidogrel-induced platelet inhibition. Fig 1 provides a simplified overview of the examined SNPs.

CYP3A4 is involved in the second oxidation step of the clopidogrel activation. The *1G allele is a gain-of-function allele resulting in increased platelet inhibition. The NR1I2 is a regulator of several enzymes including CYP2C19 and CYP3A4. A recent meta-analysis suggests that clopidogrel is primarily metabolized by CYP3A4, and that CYP2C19 is a minor pathway for clopidogrel metabolism [11].

The P2Y12 receptor binds the active clopidogrel metabolite on the platelet surface. The irreversible binding of clopidogrel to the receptor prohibits ADP from activating the platelet through this pathway and thereby decreases platelet activation. However, changes in the structure of the P2Y12 receptor caused by, for example, genetic variation in the gene encoding the receptor could potentially prevent clopidogrel from binding to the receptor. The result is less platelet inhibition and a potential phenotypic clopidogrel resistance [5].

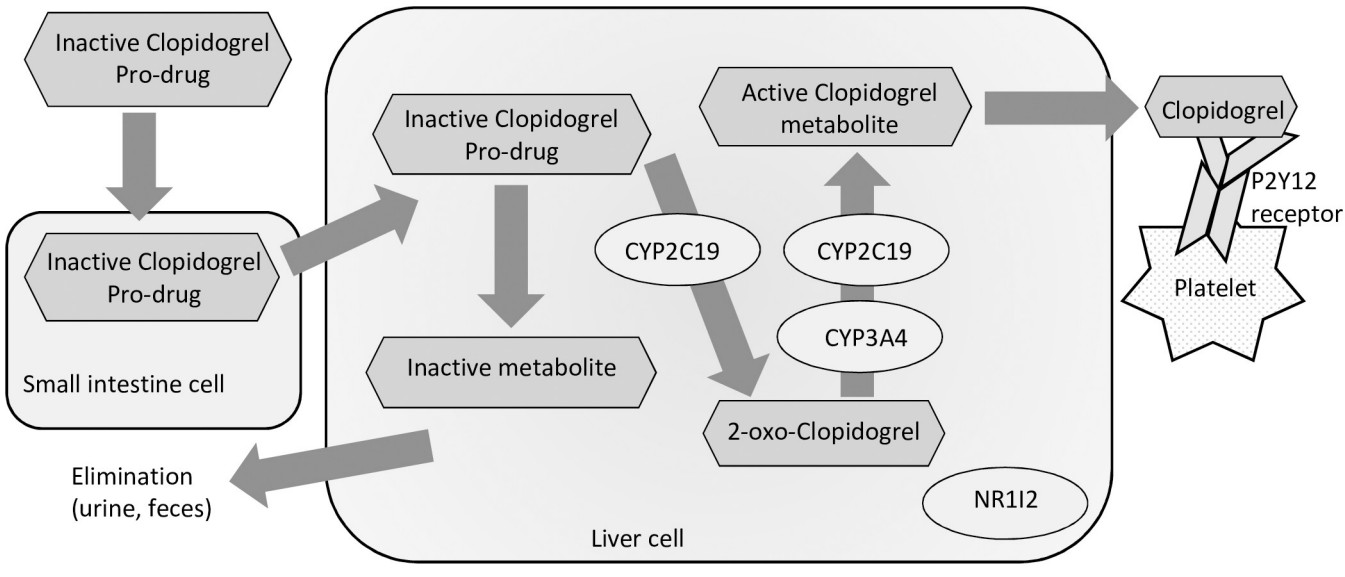

**Fig 1. Clopidogrel metabolism.**

## Materials and methods

In this observational study, we assessed patients with ischemic stroke at the Neurovascular Centre at Zealand University Hospital.

The study was approved by the local ethics committee (Den Regionale Videnskabsetiske Komite I Region Sjælland) (SJ-488) and the national competent authority (Danish Health Authority: Eudra-CT: 2015-003548-58).

Between May 2016 and August 2017, a total of 103 patients with an IS or TIA diagnosis were enrolled in the study, based on the following inclusion criteria: Clinical diagnosis of ischemic stroke or transient ischemic attack (TIA); minimum age of 18 years; secondary pro-phylaxis with 75 mg clopidogrel daily for a minimum of 5 days after a loading dose of 300 mg on day 1 or 75 mg daily for a minimum of 14 days. Exclusion criteria: Intracranial haemor-rhage; increased risk of bleeding; treatment with other antiplatelet drugs, vitamin-K antago-nists, or new oral anticoagulants (NOACs); allergy towards clopidogrel; pregnant or breastfeeding women.

Patients were seen in the outpatient clinic for platelet inhibition tests and blood sampling for genotyping. Patients were phoned prior to the visit, to encourage medication adherence. Adherence was confirmed by checking the Danish prescription database and the patient's medical history. The assessment was deferred in any patients deemed possibly non-adherent.

## Blood sampling

Blood samples were collected in buffered Na Citrate sample tubes provided by the manufac-turer (Accumetrics, San Diego, CA). The first sample was discarded. Blood sampling for HTPR-testing was performed fasting and after the patient had been resting for at least 20–30 min, by careful antecubital venepuncture by an experienced physician using a Vacutainer safety-lock system (Becton Dickinson, Franklin Lakes, NJ) with a pre-attached holder and a 21-gauge syringe. Additional blood samples were collected in EDTA-tubes and stored at -20 ˚C for later genotyping.

## Platelet function test and clopidogrel HTPR definition

The VerifyNow P2Y12 (Accumetrics, San Diego, CA) is a rapid platelet-function cartridge-based assay designed to directly measure the effects of drugs on the P2Y12 receptor. Platelet aggregation samples were analysed 10–240 min after blood sampling, using the VerifyNow P2Y12 according to the manufacturer's instructions. Results are reported in platelet reaction units (PRU), indicating adenosine diphosphate-mediated platelet aggregation specific to clopidogrel. Additionally, "Base" serves as an estimate of a patient's baseline platelet function independent of P2Y12 receptor inhibition. On the basis of a patient's Base and PRU values, the % inhibition is calculated, representing the percent change from baseline aggregation. HTPR was defined as an ex vivo PRU value ≥208 in a drug-adherent patient, measured by the VerifyNow P2Y12 assay [12, 13].

## Gene polymorphisms

Genomic deoxyribonucleic acid (DNA) was extracted from whole-blood samples using the Maxwell 16 blood DNA purification kit (Promega Corporation, Madison, WI). DNA concentrations were measured with NanoDrop 2000 (Thermo Scientific, Waltham, MA). CYP2C19*2 and CYP2C19*3 genotyping was performed by polymerase chain reaction (PCR) restriction fragment length polymorphism (RFLP) analysis. For CYP2C19*2 (rs4244285), the forward primer sequence used in PCR was 5'-AATTACAACCAGAGCTTGGC-3', and the reverse sequence 3'-AGTATAATCAATTGTCTATTATTAT-5'. For CYP2C19*3 (rs4986893), the forward primer sequence was 5'-AACATCAGGATTGTAAGCAC-3', and the reverse sequence 3'-TCAGGGCTTGGTCAATATAG-5'. The amplification protocol was 95 ˚C for 15 min, 35 cycles of 94 ˚C for 30 sec, 60 ˚C for 45 sec, and 72 ˚C for 1 min, followed by 72 ˚C for 10 min. The PCR product of CYP2C19*2 was digested and mapped using *SmaI*, and the CYP2C19*3 PCR product was digested and mapped using *BamH1*. Digested PCR products were analysed in 2% agarose gels stained with ethidium bromide.

CYP2C19*2 (rs4244285), *3 (rs4986893), and *17 (rs12248560), and CYP3A4*1G (rs2242480), NR1I2 (rs13059232), P2RY12 (rs2046934 and rs6785930), and P2Y12 (rs9859552) were simultaneously analysed with competitive allele-specific PCR (KASP; LGC Genomics, Hoddesdon, UK).

## Statistics

Power calculations were conducted with assistance from the Department of Biostatistics at Copenhagen University. We assumed that 25.7% of all patients would be carriers of at least one CYP2C19*2 allele, and 2.2% would be homozygote for the CYP2C19*2 allele [14], based on results from a study in patients with CVD [15]. This would yield a 90% chance of detecting a 5% difference between the groups with 103 included patients.

Surprisingly, however, none of the first 28 patients showed HTPR during treatment with 75 mg clopidogrel/day. Therefore, we invited patients from a previous study [16] who fulfilled the inclusion criteria and who we knew to have clopidogrel HTPR in the hyper-acute stroke phase (treated with 300 mg clopidogrel <48 hours from stroke onset, and PRU measured 8–24 hours from clopidogrel intake). Twenty-six patients agreed to participate. At the time of inclusion, they were undergoing long-term treatment with clopidogrel at a dose of 75 mg/day.

Statistical analyses were performed using IBM SPSS software for Windows, version 24 (Chicago, IL). All data, except PRU values and "days between clopidogrel initiation and PRU measurement" were normally distributed. Categorical data are expressed as frequencies and percentages, continuous data as the mean ± standard deviation (SD) or the median and interquartile ranges (IQR), as appropriate. For continuous outcome variables, differences between

groups were analysed using Student's t-tests with Levine's test for equality of variances, or the Mann-Whitney U test, as appropriate. For categorical outcome variables (all dichotomous), a chi-square test was used, in combination with Fisher's exact test when the expected values in the 2x2 tables were <5. A *p*-value <0.05 was considered statistically significant.

## Results

103 patients were examined, confirmed eligible and enrolled in the study, including the 26 patients with PRU values from the hyper-acute stroke phase who had participated in an earlier study. None showed clopidogrel HTPR with a PRU cut-off ≥208 when treated with 75 mg clopidogrel/day. Median treatment time was 69 days (IQR: 30–503) for the entire population, 492 days (IQR: 421–552) for the 26 patients with HTPR in the hyper-acute stroke phase, and 42 days (IQR: 27–114) for the remaining 77 patients. For additional patient characteristics, see Table 1.

The 26 patients with clopidogrel HTPR in the hyper-acute stroke phase all presented a similar linear PRU decline when measured under steady-state long-term treatment (Fig 2). Regardless of their PRU values in the hyper-acute phase, none of the patients exhibited clopidogrel HTPR under steady-state long-term treatment with 75 mg clopidogrel/day. They did, however, have higher PRU values on long-term treatment than the rest of the patient population (109.8 vs 81.5; p = 0.03, 95%CI:-53.5;-3.1).

The allele frequency of the examined SNPs is presented in Table 2. In 1 to 4 patients the SNP the examined SNP was undetermined, depending on the SNP in question. No patients carried the CYP2C19*3 allele. All examined SNPs were in Hardy-Weinberg equilibrium.

Of all patients, 30.7% were carriers of the CYP2C19*2 allele. Two patients were homozygote for the *2 allele (Table 2). Carriers of the *2 allele had higher PRU values (129 vs 74; p<0.01) and lower inhibition values (39 vs 65; p<0.01; 95%CI: 15.2;-36.2) (Table 3) than non-carriers. Homozygotes for the *2 allele had the highest PRU values (181.5 ± 28) and the lowest inhibition values (6 ± 2.8) (Fig 3). Even though their PRU values were higher, none of these patients exhibited HTPR.

**Table 1. Characteristics of the study population.**

|  |  | CYP2C19*2 |  | CYP2C19*17 |  | CYP23A4*1G |  | NR1I2 |  | P2Y12; rs2046934 |  | P2Y12; rs9859552 |  | P2RY12; rs6785930 |  |
|---|---|---|---|---|---|---|---|---|---|---|---|---|---|---|---|
|  | All (n = 103) | GG (n = 70) | GA/AA (n = 31) | CC (n = 71) | CT/TT (n = 31) | CC (n = 74) | CT/TT (n = 28) | TT (n = 18) | TC/CC (n = 84) | TT (n = 71) | TC/TT (n = 31) | GG (n = 67) | GT/TT (n = 35) | GG (n = 45) | GA/AA (n = 54) |
| Age, years (SD) | 67 ± 11 | 67 ± 10 | 68 ± 11 | 68 ± 10 | 69 ± 12 | 67 ± 11 | 68 ± 9 | 62 ± 12 | 69 ± 10 | 68 ± 10 | 67 ± 11 | 68 ± 11 | 67 ± 10 | 69 ± 11 | 66 ± 10 |
| Male, n (%) | 62 (60.2) | 38 (54.3) | 23 (74.2) | 44 (62.0) | 17 (54.8) | 49 (66.2) | 12 (42.9) | 10 (55.6) | 51 (60.7) | 39 (54.9) | 22 (71.0) | 40 (59.7) | 21 (60.0) | 31 (68.9) | 27 (50.0) |
| Diabetes, n (%) | 12 (11.7) | 10 (14.3) | 1 (3.2) | 6 (8.5) | 6 (19.4) | 7 (9.5) | 5 (17.9) | 3 (16.7) | 9 (10.7) | 9 (12.7) | 3 (9.7) | 8 (11.9) | 4 (11.4) | 6 (13.3) | 6 (11.1) |
| Hypertension n (%) | 48 (46.6) | 29 (41.4) | 18 (58.1) | 36 (50.7) | 12 (38.7) | 30 (40.5) | 18 (64.3) | 7 (38.9) | 41 (48.8) | 32 (45.1) | 16 (51.6) | 31 (46.3) | 17 (48.6) | 22 (48.9) | 24 (44.4) |
| Statin treatment n (%) | 91 (88.3) | 61 (87.1) | 29 (93.5) | 64 (90.1) | 26 (83.9) | 66 (89.2) | 23 (88.5) | 18 (100) | 72 (85.7) | 63 (88.7) | 27 (87.1) | 62 (92.5) | 28 (80.0) | 38 (84.4) | 49 (90.7) |
| PPI treatment n (%) | 30 (29.1) | 24 (34.3) | 5 (16.1) | 21 (29.6) | 9 (29.0) | 25 (33.8) | 5 (19.2) | 7 (38.9) | 23 (27.4) | 24 (33.8) | 6 (19.4) | 17 (25.4) | 13 (37.1) | 14 (31.1) | 15 (27.8) |
| IHD n (%) | 9 (8.7) | 5 (7.1) | 4 (12.9) | 9 (12.7) | 0 (0) | 7 (9.5) | 2 (7.1) | 0 (0) | 9 (10.7) | 7 (9.9) | 2 (6.5) | 7 (10.4) | 2 (5.7) | 7 (15.6) | 2 (3.7) |
| Ischemic stroke n (%) | 65 (63.1) | 46 (65.7) | 18 (58.1) | 45 (63.4) | 19 (61.3) | 45 (60.8) | 18 (69.2) | 14 (77.8) | 50 (59.5) | 44 (62.0) | 20 (64.5) | 40 (59.7) | 24 (68.6) | 29 (64.4) | 33 (61.1) |
| PRU (IQR) | 98 (46–130) | 74 (9–118) | 129 (83–162) | 104 (54–137) | 64 (9–126) | 103 (41–137) | 98 (60–116) | 89 (44–139) | 100 (47–130) | 103 (53–135) | 87 (9–119) | 83 (33–120) | 110 (68–139) | 110 (53–142) | 89 (41–120) |
| Inhibition % (SD) | 57 ± 28 | 65 ± 25.1 | 39 ± 23.4 | 52 ± 26.9 | 66 ± 27.2 | 57 ± 28.1 | 55 ± 25.3 | 58 ± 25.0 | 56 ± 28.2 | 56 ± 26.6 | 57 ± 30.1 | 59 ± 28.8 | 52 ± 24.8 | 50 ± 27.0 | 59 ± 27.0 |
| Baseline PRU (SD) | 204 ± 31 | 207 ± 29.5 | 198 ± 34.4 | 204 ± 33.6 | 207 ± 24.7 | 206 ± 31.6 | 198 ± 30.4 | 209 ± 40.6 | 203 ± 28.8 | 211 ± 26.4 | 189 ± 35.5 | 200 ± 32.3 | 213 ± 27.2 | 205 ± 32.9 | 204 ± 30.2 |

SD, standard deviation; PPI, proton pump inhibitor; IHD, ischemic heart disease; PRU, platelet reaction units; IQR, inter-quartile range.

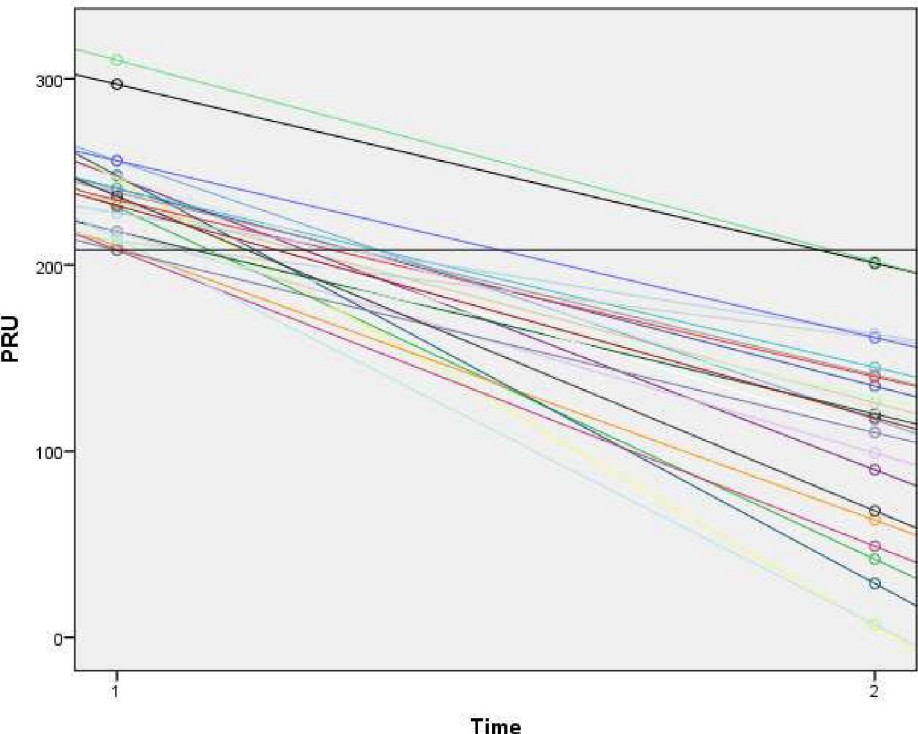

**Fig 2. PRU change over time.**

**Table 2. Allele frequency.**

| SNP | n (%) | SNP | n (%) |
|---|---|---|---|
| CYP2C19*2 | | P2Y12 rs2046934 | |
| GG | 70 (69.3) | TT | 71 (69.6) |
| GA | 29 (28.7) | TC | 25 (25.5) |
| AA | 2 (2.0) | CC | 6 (5.9) |
| Undetermined | 2 | Undetermined | 1 |
| CYP2C19*17 | | P2Y12 rs9859552 | |
| CC | 71 (69.6) | GG | 67 (65.7) |
| CT | 25 (24.5) | GT | 31 (30.4) |
| TT | 6 (5.9) | TT | 4 (3.9) |
| Undetermined | 1 | Undetermined | 1 |
| CYP23A4*16 | | P2RY12 rs6785930 | |
| CC | 74 (72.5) | GG | 45 (45.5) |
| CT | 28 (27.5) | GA | 48 (48.5) |
| TT | 0 (0) | AA | 6 (6.1) |
| Undetermined | 1 | Undetermined | 4 |
| NR1I2 | | | |
| CC | 45 (44.1) | | |
| CT | 39 (39.2) | | |
| TT | 18 (17.6) | | |
| Undetermined | 1 | | |

SNP, single nucleotide polymorphism.

**Table 3. Associations between genetic polymorpisms, platelet reactivity and % inhibition.**

| Gene | SNP | Alleles | PRU* | | | | | % inhibition** | | | | Cohens d |
|------|-----|---------|------|------|------|------|------|------|------|------|------|------|
| | | | NC median (IQR) | C median (IQR) | U statistics (z) | p-value | Effect size r | NC mean ± SD | C mean ± SD | Mean difference (95%CI) | p-value | |
| CYP2C19*2 | rs4244285 | G>A | 74 (9–118) | 129 (83–162) | 1595 (3.756) | <**0.01** | 0.37 | 65 ± 25.1 | 39 ± 23.4 | 25.7 (15.2;36.2) | <**0.01** | 1.08 |
| CYP2C19*17 | rs21248560 | C>T | 104 (54–137) | 64 (9–126) | 856 (-1.779) | 0.08 | -0.18 | 52 ± 26.9 | 66 ± 27.2 | -13.5 (-25.0;-2.0) | **0.02** | 0.53 |
| CYP23A4*16 | rs2242480 | C>T | 103 (41–137) | 98 (60–116) | 872.5 (-0.703) | 0.48 | -0.07 | 57 ± 28.1 | 55 ± 25.3 | 1.55 (-10.9;13.9) | 0.81 | 0.08 |
| NR1I2 | rs13059232 | T>C | 89 (44–139) | 100 (47–130) | 748.5 (-0.066) | 0.95 | -0.01 | 58 ± 25.0 | 56 ± 28.2 | 1.86 (-12.4; 16.1) | 0.80 | 0.08 |
| P2Y12 | rs2046934 | C>T | 103 (53–135) | 87 (9–119) | 952 (-1.081) | 0.28 | -0.11 | 56 ± 26.6 | 57 ± 30.1 | -0.95 (-12.8; 10.9) | 0.85 | 0.04 |
| P2Y12 | rs9859552 | G>T | 83 (33–120) | 110 (68–139) | 1424.5 (1.777) | 0.08 | 0.18 | 59 ± 28.8 | 52 ± 24.8 | 6.81 (-4.6; 18.2) | 0.24 | 0.26 |
| P2RY12 | rs6785930 | G>A | 110 (53–142) | 89 (41–120) | 950 (-1.863) | 0.06 | -0.19 | 50 ± 27.0 | 59 ± 27.0 | -8.86 (-19.7; -2.0) | 0.11 | 0.33 |

SNP, single nucleotide polymorphism; NC, non-carriers; C, carriers; PRU, platelet reaction units; 95%CI, 95% confidence interval; IQR, Inter quartile range. Significant p-values in bold.

*Tested with Mann-Whitney U.

** Tested with independent samples T-test.

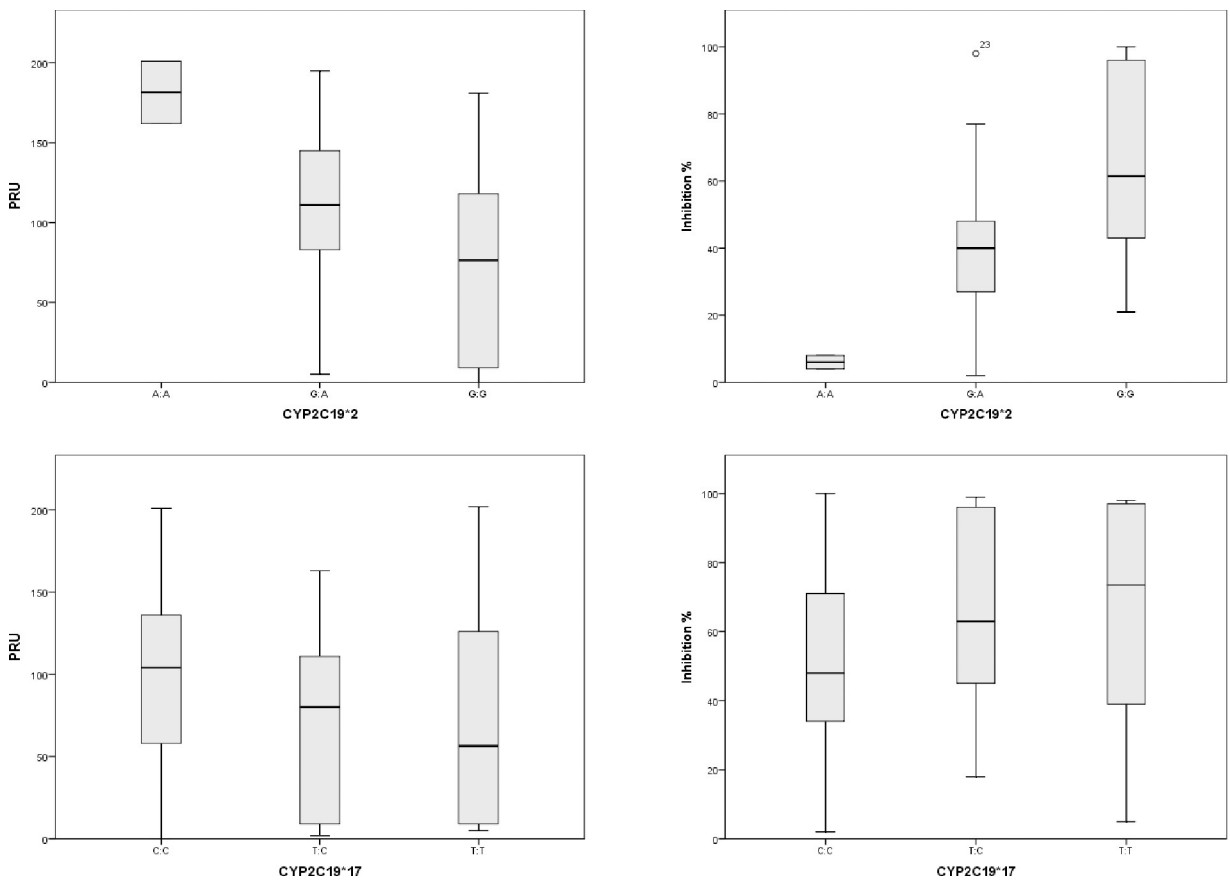

**Fig 3. PRU and inhibition.**

The CYP2C19*17 allele was found in 30.4% of patients, and six were homozygote for the *17 allele (Table 2). There was no difference in PRU between the groups (p = 0.08), but higher inhibition values were found in *17 carriers (p = 0.02; 95%CI: -25;-2) (Table 3).

We found no difference in the prevalence of *2 carriers amongst the 26 previous HTPR patients compared to the rest of the population (6/26 vs 25/77; p = 0.37).

For the remaining examined SNPs, no associations between SNPs and PRU or SNPs and % inhibition were found (Table 3).

Since no patients showed HTPR on 75 mg clopidogrel/day, the dose of clopidogrel was never increased.

## Discussion

In this observational study, we found that patients with IS and TIAs during long-term treatment with 75 mg clopidogrel/day all showed sufficient phenotypic antiplatelet responses, regardless of their genotype. Even patients who exhibited clopidogrel HTPR in the acute IS or TIA phase showed sufficient antiplatelet responses during long-term low-dose clopidogrel treatment.

The results from the 26 patients with PRU values from the hyper-acute phase indicate that HTPR status shifts with continued medical therapy. Furthermore, even though these patients show higher PRU values during long-term treatment than the rest of the patient population, they are not high enough to determine HTPR. Studies in patients with CVD have found that HTPR status can shift over time, not only from HTPR to non-HTPR, but also vice versa [17]. Other studies report results similar to ours, namely that with continued treatment, some patients with HTPR shifted to non-HTPR [12, 18, 19]. HTPR in the acute stroke phase might be explained by platelet hyper-reactivity [20, 21]. A combination of increased platelet reactivity in the acute stroke phase and some patients taking longer than 7–10 days to achieve full platelet inhibition might explain the lack of patients with HTPR in our study, since the median clopidogrel treatment time was 69 days (IQR: 30–503). However, our conclusion from the present study is that patients with IS or TIAs who receive long-term clopidogrel treatment at a dose of 75 mg/day have sufficient antiplatelet responses.

Carriers of the CYP2C19*2 LOF allele showed higher platelet reactivity (higher PRU values) than non-carriers, as well as higher PRU values with increasing numbers of the CYP2C19*2 allele. However, their values were still not high enough to determine HTPR. We also found that carriers of the CYP2C19*17 GOF allele showed increased platelet inhibition compared to non-carriers.

While several studies have examined the associations between carrying specific SNPs, ex vivo platelet function, and the risk of recurrent stroke or vascular events, the results are conflicting. A meta-analysis by Pan et al. published in January 2017 [9] concluded that carriers of CYP2C19 LOF SNPs have a higher risk of stroke and vascular events than non-carriers among clopidogrel-treated patients with previous strokes or transient ischemic attacks. In this meta-analysis, platelet function was not explored. However, in a study by Han et al. [6], CYP2C19 LOF SNPs had a significant impact on platelet responses, but were not associated with clinical endpoints (ischemic events and bleeding). Yet again, Fiolaki et al. [3] found an increased risk of recurrent IS or TIAs in patients with HTPR. In their analysis, SNPs were not examined.

The majority of the studies are conducted in Asian populations that have been shown to differ genetically from white populations. The prevalence of CYP2C19*2 carriers is higher in Asian than in white populations, while the CYP2C19*17 allele is rare in Asian but frequent in white populations [22]. The frequencies of CYP2C19*2 and CYP2C19*17 in our study were 30.7% and 30.4%, respectively, which is much higher than what the Clinical Pharmacogenetics Implementation

Consortium (CPIC) guidelines for CYP2C19 genotype and clopidogrel therapy in white patients indicate [22], but are in line with previous studies in patients with CVD and stroke [23, 24]. Despite 30.7% of patients carrying the *2 LOF allele, none showed clopidogrel HTPR.

One might have expected the prevalence of *2 to be higher in the 26 patients with PRU values from the hyper-acute phase, as an explanation for their acute state and the persistently higher PRU values, but this was not the case. The distribution of the CYP2C19*2 allele in the 26 patients with acute-phase HTPR did not differ from the rest of the population in our study. We did find that long-term PRU values increase with the number of *2 alleles, but there does not seem to be an association with acute-phase PRU values. However, acute-phase PRU values seem to predict persistently higher PRU values. We acknowledge that we did not have any patient who we knew to have non-HTPR in the acute phase for comparison. Similarly, previous studies have found no association between the CYP2C19*2 allele and HTPR [25], and the clinical relevance of the *2 allele has been questioned [11, 26].

Several studies have investigated the effects of different SNPs on the clopidogrel metabolism, and apart from the CYP2C19*2 allele, there is no evidence that other SNPs alone play any role in clopidogrel resistance. However, it has been suggested that the combination of several SNPs is important, rather than the effect of any SNP alone [27].

The present study has several limitations. The small sample size prevented us from conducting a sub-group analysis on selected SNPs as well as on the interactions between SNPs. A post-hoc effect size analysis resulted in very small to small effect sizes (Cohens *d*: *0.01–0.4*) for all SNPs except for CYP2C19*2. The relatively small sample size might also be the reason that we did not find any patients with HTPR.

One could argue that we should have chosen a different way of determining HTPR. We have used the widely accepted dichotomization of the PRU value. We have even set the value at 208, based on the results from the GRAVITAS trial, to increase sensitivity [28]. In spite of that, we did not find any patients with HTPR. Another way to determine HTPR is to look for platelet inhibition >20% or use a longitudinal definition of HTPR, as proposed by Tobin [29]. This would however have required a completely different study design, with blood samples collected prior to treatment initiation.

We cannot rule out selection bias in our study, since the patients who are most disabled from their stroke will hesitate to participate in a study that requires traveling from their home or care facility to a hospital. Hence, patients with TIAs or minor stroke are more likely to participate in such studies.

The lack of a clinical end point leaves us unable to determine if the most important factor in the risk of recurrent stroke or vascular events are different SNPs or phenotypic platelet resistance.

In conclusion, we find that patients with ischemic stroke undergoing long-term treatment with 75 mg clopidogrel/day do not have HTPR. Even in a group of patients with HTPR in the acute stroke phase, we did not find any patients with HTPR. Carriers of the CYP2C19*2 allele did have higher PRU values than non-carriers, but their values were not high enough to determine HTPR. In conclusion, a genetic analysis of CYP2C19*2 in white patients seems futile in determining the patient's long-term treatment clopidogrel resistance. Acute-phase PRU values seem to predict persistently higher PRU values. An adequately powered observational study with a long-term follow-up and clinical outcomes assessing both genetic differences and, repeatedly, measures of platelet inhibition in patients with ischemic stroke is much needed.

## Supporting information

**S1 Data.**
(DOCX)

## Author Contributions

**Conceptualization:** Charlotte Lützhøft Rath, Niklas Rye Jørgensen, Troels Wienecke.

**Data curation:** Charlotte Lützhøft Rath, Troels Wienecke.

**Formal analysis:** Charlotte Lützhøft Rath, Niklas Rye Jørgensen, Troels Wienecke.

**Investigation:** Charlotte Lützhøft Rath.

**Methodology:** Charlotte Lützhøft Rath, Niklas Rye Jørgensen, Troels Wienecke.

**Project administration:** Charlotte Lützhøft Rath, Niklas Rye Jørgensen, Troels Wienecke.

**Resources:** Charlotte Lützhøft Rath, Niklas Rye Jørgensen, Troels Wienecke.

**Software:** Charlotte Lützhøft Rath.

**Supervision:** Niklas Rye Jørgensen, Troels Wienecke.

**Validation:** Charlotte Lützhøft Rath, Niklas Rye Jørgensen, Troels Wienecke.

**Visualization:** Charlotte Lützhøft Rath, Niklas Rye Jørgensen, Troels Wienecke.

**Writing – original draft:** Charlotte Lützhøft Rath.

**Writing – review & editing:** Charlotte Lützhøft Rath, Niklas Rye Jørgensen, Troels Wienecke.

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
