## [Decision Letter · Decision Letter 0]

10 Dec 2019

PONE-D-19-31152

CLOPIDOGREL RESPONDER STATUS IS UNINFLUENCED BY CYP2C19*2 IN DANISH PATIENTS WITH STROKE

PLOS ONE

Dear Ms Rath,

Thank you for submitting your manuscript to PLOS ONE. After careful consideration, we feel that it has merit but does not fully meet PLOS ONE’s publication criteria as it currently stands. Therefore, we invite you to submit a revised version of the manuscript that addresses the points raised during the review process.

The manuscript has been reviewed by two experts in the field. They find a series of specific-minor comments that have to be considered by the authors, including discussing some important studies in the field. Of more importance is the discussion brought by one of the reviewers on the prevalence and determination of high on treatment platelet reactivity in the studied group. It seems to be at odds with other studies in the field. The recomendations of the reviewer should be consdered, including a re-analysis of the results with the suggested cut-off values for PRU.

We would appreciate receiving your revised manuscript by Jan 24 2020 11:59PM. To enhance the reproducibility of your results, we recommend that if applicable you deposit your laboratory protocols in protocols.io, where a protocol can be assigned its own identifier (DOI) such that it can be cited independently in the future. For instructions see: http://journals.plos.org/plosone/s/submission-guidelines#loc-laboratory-protocols

We look forward to receiving your revised manuscript.

Kind regards,

Pablo Garcia de Frutos

Academic Editor

PLOS ONE

Journal Requirements:

2. Please note that according to our submission guidelines (http://journals.plos.org/plosone/s/submission-guidelines), outmoded terms and potentially stigmatizing labels should be changed to more current, acceptable terminology. For example: “Caucasian” should be changed to “white” or “of [Western] European descent” (as appropriate).

3.  We noticed you have some minor occurrence of overlapping text with the following previous work, which needs to be addressed:

https://doi.org/10.1016/j.jstrokecerebrovasdis.2018.05.027

In your revision ensure you cite all your sources (including your own works), and quote or rephrase any duplicated text outside the methods section. Further consideration is dependent on these concerns being addressed.

4. Thank you for including your ethics statement: "The study was approved by the local ethics committee (SJ-488) and the national competent authority (Danish Health Authority: Eudra-CT: 2015-003548-58). all patients gave oral and wrtitten informed consent."

Reviewers' comments:

Reviewer's Responses to Questions

**Comments to the Author**

1. Is the manuscript technically sound, and do the data support the conclusions?

Reviewer #1: No

Reviewer #2: Yes

2. Has the statistical analysis been performed appropriately and rigorously? 

Reviewer #1: Yes

Reviewer #2: Yes

3. Have the authors made all data underlying the findings in their manuscript fully available?

Reviewer #1: Yes

Reviewer #2: Yes

4. Is the manuscript presented in an intelligible fashion and written in standard English?

Reviewer #1: Yes

Reviewer #2: Yes

5. Review Comments to the Author

Reviewer #1: It is a pharmacogenetic study that evaluates the influence of CYP2C19, CYP3A4, NR1I2 and PR2Y12 polymorphisms in the response to clopidogrel in patients with a history of stroke or transient ischemic attacks.

As expect, they found a relationship between CYP2C19 polymorphisms and platelet reaction unit (PRU) but no relation with the other genes. Surprisingly, they did not find a relation of CYP2C19 polymorphisms with high-on-treatment platelet reactivity (HTPR) to clopidogrel, a way to measure no-response to this drug. This can be explain by a methodological error in the measurement of PRU because the incidence of HTPR in other studies varies from 8 to 61% (mean 27%) and the prevalence in this study is 0%. How can we be sure that PRU results are reliable? If there are no HTPR patients, it is impossible to find an association with any factor.

Moreover, HTPR was defined as an PRU value >208 and other studies use a cut-off of >185 (see Saiz-Rodríguez et al. Influence of CYP450 Enzymes, CES1, PON1, ABCB1, and P2RY12 Polymorphisms on Clopidogrel Response in Patients Subjected to a Percutaneous Neurointervention. Clin Ther. 2019 Jun;41(6):1199-1212).

On the other hand, the influence of concomitant medication, as proton pump inhibitors (PPI), that inhibit CYP2C19 has not been evaluated. More CYP2C19 wild type patients are receiving PPI (34.3%) than CYP2C19*2 carriers (16.1%) and this difference can influence the comparison of the two groups.

Reviewer #2: Abstract: HTPR: is not explained in the abstract

Introduction: The aim of the study sholud be mention in the Intruduction section.

Please indicated the reasosns for investigated the named SNPs: CYP2C19*2 (rs4244285), *3 (rs4986893), and *17 (rs12248560), and CYP3A4*1G (rs2242480).

NR1I2 (rs13059232), P2RY12 (rs2046934 and rs6785930), and P2Y12 (rs9859552)

Method: The authors should explainne why they use PCR_RFLP mathod and competitive allele-specific PCR (KASP) for genotyping of CYP2C19*2 (rs4244285), *3 (rs4986893) SNPs? By KASP they may genotyped also competitive allele-specific PCR (KASP). I do not understand why they used also PCR-RFLP method

Why the authors did not exclude the cases with undetermined SNPs.?

In Results section: Please check and reformulate „ In 1-4 patients the SNP the examined SNP was undetermined.” It is not clear..... Patient number 10 (from Supplemental material has only two SNPs determined.and 6 undetermined. Why he was not excluded from the study ?

The authors should discus also another study performe in europeans patients (189 clopidogrel-treated patients with acute coronary syndromes and noncardiogenic ischemic stroke).

Mărginean A, et al. The Impact of CYP2C19 Loss-of-Function Polymorphisms, Clinical, and Demographic Variables on Platelet Response to Clopidogrel Evaluated Using Impedance Aggregometry. Clin Appl Thromb Hemost. 2017 Apr;23(3):255-265. doi: 10.1177/1076029616629211. Epub 2016 Jul 9

6. PLOS authors have the option to publish the peer review history of their article (what does this mean?). If published, this will include your full peer review and any attached files.

Reviewer #1: Yes: Francisco Abad-Santos

Reviewer #2: No

---

## [Author Response · Author response to Decision Letter 0]

24 Jan 2020

Dear reviewers.

Thank you for your thorough review. Please see attached file for detailed response.

---

## [Decision Letter · Decision Letter 1]

6 Feb 2020

PONE-D-19-31152R1

CLOPIDOGREL RESPONDER STATUS IS UNINFLUENCED BY CYP2C19*2 IN DANISH PATIENTS WITH STROKE

PLOS ONE

Dear Ms Rath,

Thank you for submitting your manuscript to PLOS ONE. After careful consideration, we feel that it has merit but does not fully meet PLOS ONE’s publication criteria as it currently stands. Therefore, we invite you to submit a revised version of the manuscript that addresses the points raised during the review process.

The study has been evaluated by one reviewer, that considers that there are two issues remaining in the present version. The authors should respond to these issues by analyzing the influence of PPI treatment and discussing the possible cause of the low incidence of HTPR in their study compared to other studies.

We would appreciate receiving your revised manuscript by Mar 22 2020 11:59PM. To enhance the reproducibility of your results, we recommend that if applicable you deposit your laboratory protocols in protocols.io, where a protocol can be assigned its own identifier (DOI) such that it can be cited independently in the future. For instructions see: http://journals.plos.org/plosone/s/submission-guidelines#loc-laboratory-protocols

We look forward to receiving your revised manuscript.

Kind regards,

Pablo Garcia de Frutos

Academic Editor

PLOS ONE

Reviewers' comments:

Reviewer's Responses to Questions

**Comments to the Author**

1. If the authors have adequately addressed your comments raised in a previous round of review and you feel that this manuscript is now acceptable for publication, you may indicate that here to bypass the “Comments to the Author” section, enter your conflict of interest statement in the “Confidential to Editor” section, and submit your "Accept" recommendation.

Reviewer #1: (No Response)

2. Is the manuscript technically sound, and do the data support the conclusions?

Reviewer #1: Partly

3. Has the statistical analysis been performed appropriately and rigorously? 

Reviewer #1: Yes

4. Have the authors made all data underlying the findings in their manuscript fully available?

Reviewer #1: Yes

5. Is the manuscript presented in an intelligible fashion and written in standard English?

Reviewer #1: Yes

6. Review Comments to the Author

Reviewer #1: There are several questions that have not been answered:

1. The authors did not find any patient with high-on-treatment platelet reactivity (HTPR) to clopidogrel, when the incidence of HTPR in other studies varies from 8 to 61% (mean 27%). How can we be sure that platelet reaction unit (PRU) results are reliable?

2. They justify that they measured PRU values not in the acute phase (at least 14 days after stroke or TIA) which is probably why they have not found any patients with HTPR. This should be commented in the discussion and some references should be included to support this argument.

3. The influence of PPI treatment should be statistically analysed as well as their interaction with CYP2C19 polymorphisms; this information should be included in results and commented in discussion.

7. PLOS authors have the option to publish the peer review history of their article (what does this mean?). If published, this will include your full peer review and any attached files.

Reviewer #1: No

---

## [Author Response · Author response to Decision Letter 1]

26 Jun 2020

Response to reviewers.

Please see responses below. We hope you find them satisfactional.

Reviewer #1:

1) The authors did not find any patient with high-on-treatment platelet reactivity (HTPR) to clopidogrel, when the incidence of HTPR in other studies varies from 8 to 61% (mean 27%). How can we be sure that platelet reaction unit (PRU) results are reliable?

Answer: We have followed all instructions from the manufacturer regarding blood sampling and analysis. All blood sampling and analysis were done by staff familiar with the procedures.

2) They justify that they measured PRU values not in the acute phase (at least 14 days after stroke or TIA) which is probably why they have not found any patients with HTPR. This should be commented in the discussion and some references should be included to support this argument.

Answer: We believe this is already addressed. Please see yellow highlighted section in the discussion.

3) The influence of PPI treatment should be statistically analysed as well as their interaction with CYP2C19 polymorphisms; this information should be included in results and commented in discussion

Answer: The aim of this study is not to investigate whether different PPI´s affect clopidogrelresponse. The study is not sufficiently powered to investigate different types of PPI´s. We are not aware of any study that has found different PPI´s in patients treated with clopidogrel to have an effect on clinical outcome.

---

## [Decision Letter · Decision Letter 2]

6 Jul 2020

CLOPIDOGREL RESPONDER STATUS IS UNINFLUENCED BY CYP2C19*2 IN DANISH PATIENTS WITH STROKE

PONE-D-19-31152R2

Dear Dr. Rath,

We’re pleased to inform you that your manuscript has been judged scientifically suitable for publication and will be formally accepted for publication once it meets all outstanding technical requirements.

Kind regards,

Pablo Garcia de Frutos

Academic Editor

PLOS ONE

Additional Editor Comments (optional):

Reviewers' comments:

Reviewer's Responses to Questions

**Comments to the Author**

1. If the authors have adequately addressed your comments raised in a previous round of review and you feel that this manuscript is now acceptable for publication, you may indicate that here to bypass the “Comments to the Author” section, enter your conflict of interest statement in the “Confidential to Editor” section, and submit your "Accept" recommendation.

Reviewer #1: (No Response)

2. Is the manuscript technically sound, and do the data support the conclusions?

Reviewer #1: Yes

3. Has the statistical analysis been performed appropriately and rigorously? 

Reviewer #1: Yes

4. Have the authors made all data underlying the findings in their manuscript fully available?

Reviewer #1: Yes

5. Is the manuscript presented in an intelligible fashion and written in standard English?

Reviewer #1: Yes

6. Review Comments to the Author

Reviewer #1: Although the aim of this study is not to investigate whether different PPIs affect clopidogrel response, it would be interesting to evaluate the influence of PPI treatment as well as their interaction with CYP2C19 polymorphisms. There is a previous study (Saiz-Rodríguez et al. Influence of CYP450 Enzymes, CES1, PON1, ABCB1, and P2RY12 Polymorphisms on Clopidogrel Response in Patients Subjected to a Percutaneous Neurointervention. Clin Ther. 2019;41(6):1199-1212.e2) that evaluates the influence of different PPIs in patients treated with clopidogrel to have an effect on clinical outcome.

7. PLOS authors have the option to publish the peer review history of their article (what does this mean?). If published, this will include your full peer review and any attached files.

Reviewer #1: No

---

## [Editor Report · Acceptance letter]

15 Jul 2020

PONE-D-19-31152R2 

CLOPIDOGREL RESPONDER STATUS IS UNINFLUENCED BY CYP2C19*2 IN DANISH PATIENTS WITH STROKE 

Dear Dr. Rath:

I'm pleased to inform you that your manuscript has been deemed suitable for publication in PLOS ONE. Congratulations! Your manuscript is now with our production department. 

Kind regards, 

on behalf of

Dr. Pablo Garcia de Frutos 

Academic Editor

PLOS ONE